# UHRF1 is essential for proper cytoplasmic architecture and function of mouse oocytes and derived embryos

Shuhei Uemura[1],*, Shoji Maenohara[1,2],*, Kimiko Inoue[3], Narumi Ogonuki[3], Shogo Matoba[3], Atsuo Ogura[3], Mayuko Kurumizaka[4], Kazuo Yamagata[4,5], Jafar Sharif[6], Haruhiko Koseki[6], Koji Ueda[7], Motoko Unoki[1,8], Hiroyuki Sasaki[1]

Ubiquitin-like with PHD and RING finger domains 1 (UHRF1) is a protein essential for the maintenance of DNA methylation in somatic cells. However, UHRF1 is predominantly localized in the cytoplasm of mouse oocytes and preimplantation embryos, where it may play a role unrelated to the nuclear function. We herein report that oocyte-specific *Uhrf1* KO results in impaired chromosome segregation, abnormal cleavage division, and pre-implantation lethality of derived embryos. Our nuclear transfer experiment showed that the phenotype is attributable to cytoplasmic rather than nuclear defects of the zygotes. A proteomic analysis of KO oocytes revealed the down-regulation of proteins associated with microtubules including tubulins, which occurred independently of transcriptomic changes. Intriguingly, cytoplasmic lattices were disorganized, and mitochondria, endoplasmic reticulum, and components of the subcortical maternal complex were mislocalized. Thus, maternal UHRF1 regulates the proper cytoplasmic architecture and function of oocytes and preimplantation embryos, likely through a mechanism unrelated to DNA methylation.

## Introduction

Ubiquitin-like with plant homeodomain (PHD) and really interesting new gene (RING) finger domains 1 (UHRF1) is a protein essential for the maintenance and propagation of DNA methylation patterns through DNA replication in somatic cells (Bostick et al, 2007; Sharif et al, 2007; Unoki & Sasaki 2022). It contains a ubiquitin-like (UBL) domain, a tandem Tudor domain (TTD), a PHD, a SET- and RING-associated (SRA) domain, and a RING finger domain.

The SRA domain has a strong binding affinity to hemi-methylated DNA (Arita et al, 2008; Avvakumov et al, 2008; Hashimoto et al, 2008), consistent with the UHRF1's role in the maintenance of methylation. The RING domain mediates mono-ubiquitylation of histone H3 and proliferating cell nuclear antigen–associated factor 15 (PAF15) via its E3 ligase activity, and the ubiquitylated proteins recruit the maintenance DNA methyltransferase 1 (DNMT1) (Nishiyama et al, 2013, 2020). The UBL domain interacts with an E2 ubiquitin–conjugating enzyme and coordinates the structure of UHRF1 for ubiquitylation (Citterio et al, 2004; Jenkins et al, 2005; Nishiyama et al, 2013, 2020; DaRosa et al, 2018; Foster et al, 2018). It also interacts with DNMT1 via its UBL and SRA domains (Berkyurek et al, 2014; Li et al, 2018a). The TTD recognizes histone H3 di-/tri-methylated at lysine 9 (H3K9me2/3) and DNA ligase 1 di-/tri-methylated at lysine 126 (LIG1K126me2/3) (Karagianni et al, 2008; Ferry et al, 2017), and the PHD recognizes unmodified N-termini of histone H3 and PAF15 (Arita et al, 2012; Nishiyama et al, 2020). Thus, UHRF1 interacts with various molecules via the distinctive domains for coordinated maintenance methylation.

We and others previously reported that UHRF1 is predominantly present in the cytoplasm of mouse oocytes and preimplantation embryos, with only a small proportion of the protein found in the nucleus (Maenohara et al, 2017; Li et al, 2018b; Cao et al, 2019). It was also reported that maternal UHRF1 is essential for preimplantation development: most heterozygous embryos derived from *Uhrf1* KO oocytes fertilized with WT sperm (maternal KO or mat-KO embryos) died before implantation (Maenohara et al, 2017; Cao et al, 2019). This phenotype is much more severe than that observed for *Dnmt1* mat-KO embryos, which typically die after embryonic day 14.0 (E14.0) (Howell et al, 2001). In addition, *Dnmt3l* mat-KO embryos, which are derived from oocytes with very low DNA methylation (Shirane et al, 2013), survive beyond implantation but die around E10.0 (Bourc'his et al, 2001; Hata et al, 2002). Thus, despite the

---

[1]Division of Epigenomics and Development, Medical Institute of Bioregulation, Kyushu University, Fukuoka, Japan   [2]Department of Obstetrics and Gynecology, Graduate School of Medical Sciences, Kyushu University, Fukuoka, Japan   [3]Bioresource Engineering Division, RIKEN BioResource Research Center (BRC), Ibaraki, Japan   [4]Center for Genetic Analysis of Biological Responses, Research Institute for Microbial Diseases, Osaka University, Osaka, Japan   [5]Faculty of Biology-Oriented Science and Technology, KINDAI University, Wakayama, Japan   [6]Laboratory for Developmental Genetics, RIKEN Center for Integrative Medical Sciences, Yokohama, Japan   [7]Cancer Proteomics Group, Cancer Precision Medicine Center, Japanese Foundation for Cancer Research, Tokyo, Japan   [8]Department of Human Genetics, School of International Health, Graduate School of Medicine, The University of Tokyo, Tokyo, Japan

Correspondence: hsasaki@bioreg.kyushu-u.ac.jp; unokim@m.u-tokyo.ac.jp
*Shuhei Uemura and Shoji Maenohara contributed equally to this work

---

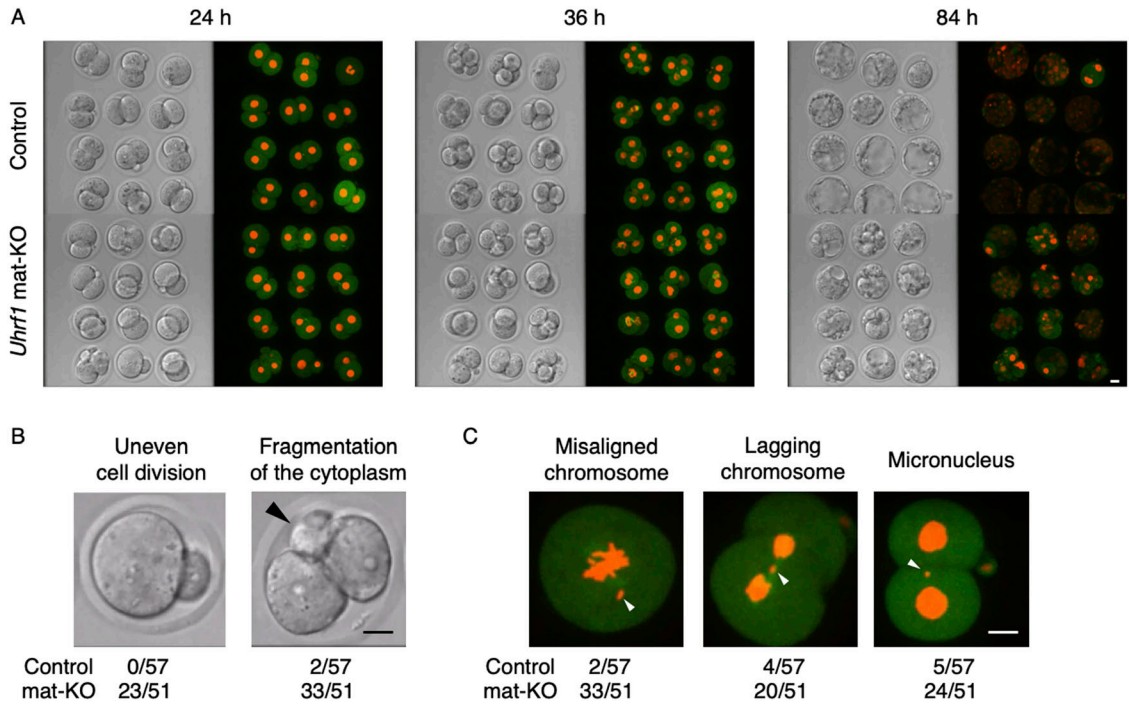

**Figure 1. Impaired chromosome segregation and cleavage division in *Uhrf1* mat-KO embryos.**
**(A)** Snapshots of live-cell imaging of control [*Uhrf1*^2lox/+] and *Uhrf1* mat-KO embryos (n = 12 each; see also Video 1). Both bright-field (left) and fluorescent images (right) are shown. mCherry-tagged histone H2B (red) and EGFP-tagged α-tubulin (green) were used as a nuclear and a cytoplasmic marker, respectively. Scale bar, 20 μm. **(B)** Representative images of abnormal cleavage division of *Uhrf1* mat-KO embryos. An empty blastomere with no nucleus is indicated by an arrowhead. Embryos with each defect were counted in five independent movies, and their total number per observed embryo is indicated (n = 57 and 51 for control and *Uhrf1* mat-KO, respectively) below the image. Scale bar, 20 μm. **(C)** Representative images of abnormal chromosome segregation in *Uhrf1* mat-KO embryos. Arrowheads indicate an example of a misaligned chromosome (left), lagging chromosome (middle), and micronucleus (right). The number of defective embryos per observed embryo is indicated below each image. Scale bar, 20 μm.

presence of DNA methylation defects in *Uhrf1* mat-KO embryos (Maenohara et al, 2017), the major cause of the preimplantation lethality is likely something else.

In the present study, we report that mouse *Uhrf1* KO oocytes and mat-KO embryos have defects in the cytoplasmic architecture and function and that they play a role in the preimplantation lethality. We also find alterations in the amount of certain cytoplasmic proteins occurring independent of the transcriptomic changes. Our results suggest a previously unknown role of maternal UHRF1 in oocytes and preimplantation embryos.

## Results

### Oocyte-specific *Uhrf1* KO impairs chromosome segregation, cleavage division, and the cell structure and function

We obtained cumulus–oocyte complexes containing metaphase II (MII) oocytes from the ovaries of oocyte-specific *Uhrf1* KO females carrying a *zona pellucida glycoprotein 3* (*Zp3*)-Cre transgene [*Uhrf1*^2lox/2lox, *Zp3*-Cre] and fertilized them with WT sperm to derive *Uhrf1* mat-KO [*Uhrf1*^1lox/+] zygotes (Maenohara et al, 2017). Previous studies reported that *Uhrf1* mat-KO embryos show preimplantation lethality, with only roughly 20% reaching the blastocyst stage

(Maenohara et al, 2017; Cao et al, 2019). Live-cell imaging of mat-KO embryos expressing mCherry-tagged histone H2B (nuclear marker) and EGFP-tagged α-tubulin (cytoplasmic marker) revealed that a great majority of the embryos were arrested by the morula stage with aberrant chromosome segregation and cell division (Fig 1A and Video 1). The defects included uneven cell division, cytoplasmic fragmentation, misaligned chromosomes, lagging chromosomes, and micronucleus formation (Fig 1B and C). Some of the embryos already showed such abnormalities at the first cell division. These defects are consistent with the previously reported abnormal spindle formation and aneuploidy in *Uhrf1* KO MII oocytes, as well as the increased accumulation of double-strand breaks in fully grown oocytes (FGOs) and mat-KO two-cell embryos (Cao et al, 2019).

In addition to the defects in cell division and chromosome segregation, Cao et al (2019) reported that *Uhrf1* KO MII oocytes have a larger perivitelline space than control MII oocytes. We realized that upon nuclear transfer using micropipettes (see next section), mat-KO zygotes have reduced surface tension and a flabbier cell body than control [*Uhrf1*^2lox/+] zygotes. Live-cell imaging showed the extraordinarily radical movement of cytoplasmic granules in mat-KO zygotes (Video 2). These findings suggest that *Uhrf1* KO oocytes and mat-KO embryos have defects in not only chromosome segregation and cleavage division but also the cytoplasmic architecture and function; in fact, the former defects may be due to the latter.

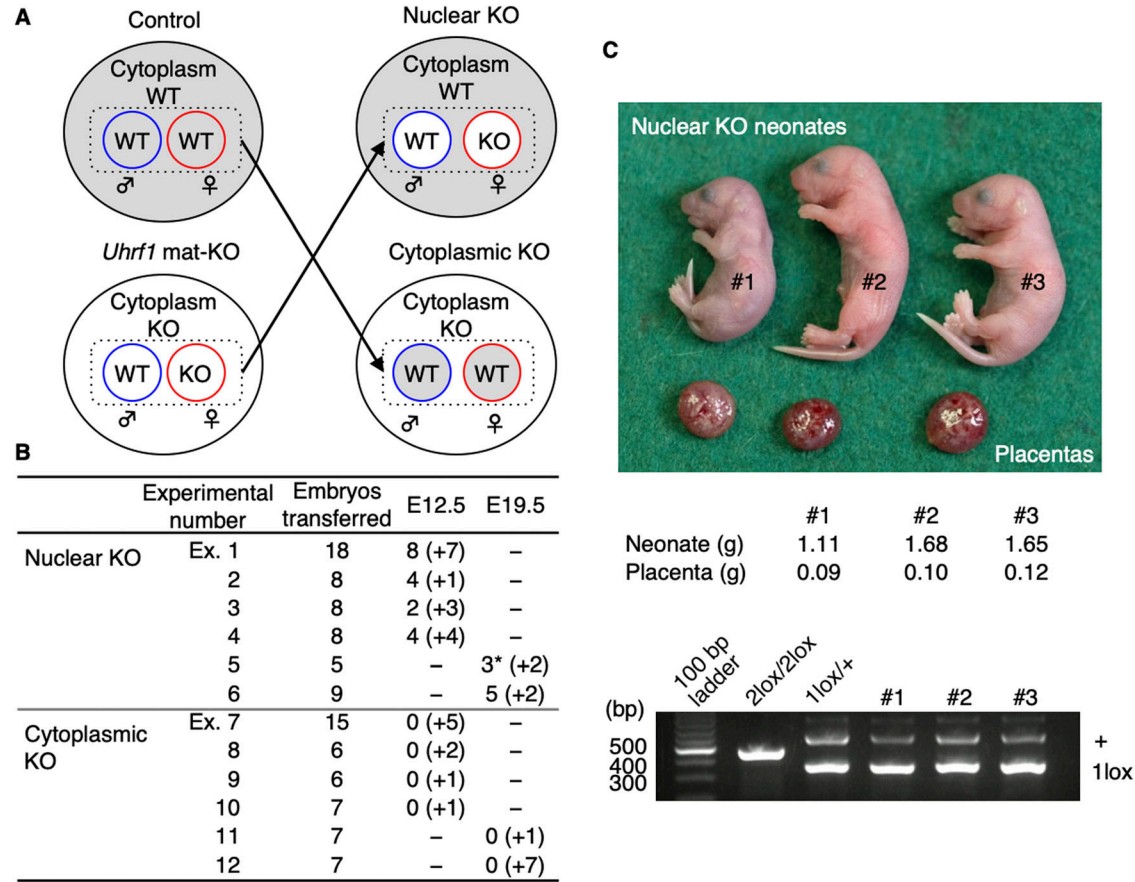

**Figure 2. Lethality of *Uhrf1* mat-KO embryos is attributable to cytoplasmic defects.**
**(A)** Production of nuclear KO and cytoplasmic KO zygotes by exchanging pronuclei between control [*Uhrf1*^2lox/+^] and mat-KO zygotes using micropipettes. Reconstructed two-cell embryos were transferred to the oviducts of pseudopregnant females and recovered at E12.5 or E19.5 (Caesarean section). **(B)** Summary of the developmental capacity of the reconstructed zygotes. Six embryo transfer experiments were done for each pronucleus/cytoplasm constitution. The number indicated in parentheses represents resorption detected upon recovery. **(C)** Image of the three live nuclear KO neonates indicated with an asterisk in (B) and their placentae (top). Their weights (middle) and genotyping results (bottom) are also shown.

## Lethality of mat-KO embryos is attributable to cytoplasmic defects

UHRF1 protein is present in both the cytoplasm and the nucleus at the oocyte and preimplantation stages (Maenohara et al, 2017; Cao et al, 2019). To determine whether the cause of the lethality resides in the cytoplasm or nucleus of *Uhrf1* mat-KO embryos, we produced "nuclear KO zygotes" with the pronuclei from KO zygotes and the cytoplasm from control [*Uhrf1*^2lox/+^] zygotes, and "cytoplasmic KO zygotes" with a reciprocal pronucleus/cytoplasm constitution by nuclear transfer (Fig 2A). Reconstructed embryos were then transferred to the oviducts of pseudopregnant females and allowed to develop to E12.5. As a result, although 42.9% (n = 18/42; four embryo transfers) of the nuclear KO embryos developed normally, none (n = 0/34; four embryo transfers) of the cytoplasmic KO embryos did (Fig 2B). Reconstructed embryos of the second cohort were allowed to develop further and recovered by Caesarean section at E19.5. Although 57.1% (n = 8/14; two embryo transfers) of the nuclear KO embryos were born alive, none (n = 0/14; two embryo transfers) of the cytoplasmic KO embryos were recovered

(Experiment 2) (Fig 2B). Of the three nuclear KO pups obtained from one foster mother (Fig 2C), two (females) died before 12 wk old, but one (male) survived beyond 18 wk and gave rise to healthy pups itself. Five nuclear KO pups (three females and two males) obtained from the other female survived at least until weaning. These results suggest that *Uhrf1* mat-KO pronuclei can support embryonic development and that the major cause of the preimplantation lethality resides in the cytoplasm. This in turn suggests that UHRF1 in oocytes is important for the formation of developmentally capable cytoplasm in zygotes.

## UHRF1 is required for proper regulation of proteins associated with cytoskeletal organization

To clarify the molecular defect underlying the phenotype of *Uhrf1* KO oocytes and mat-KO embryos, we investigated the effect of UHRF1 depletion on the transcriptome and proteome of FGOs. Depletion had no effect on the total amount of nucleic acid (mostly RNA) or protein in FGOs (Fig 3A and B), suggesting that any changes, if present, were gene/protein-specific. We therefore

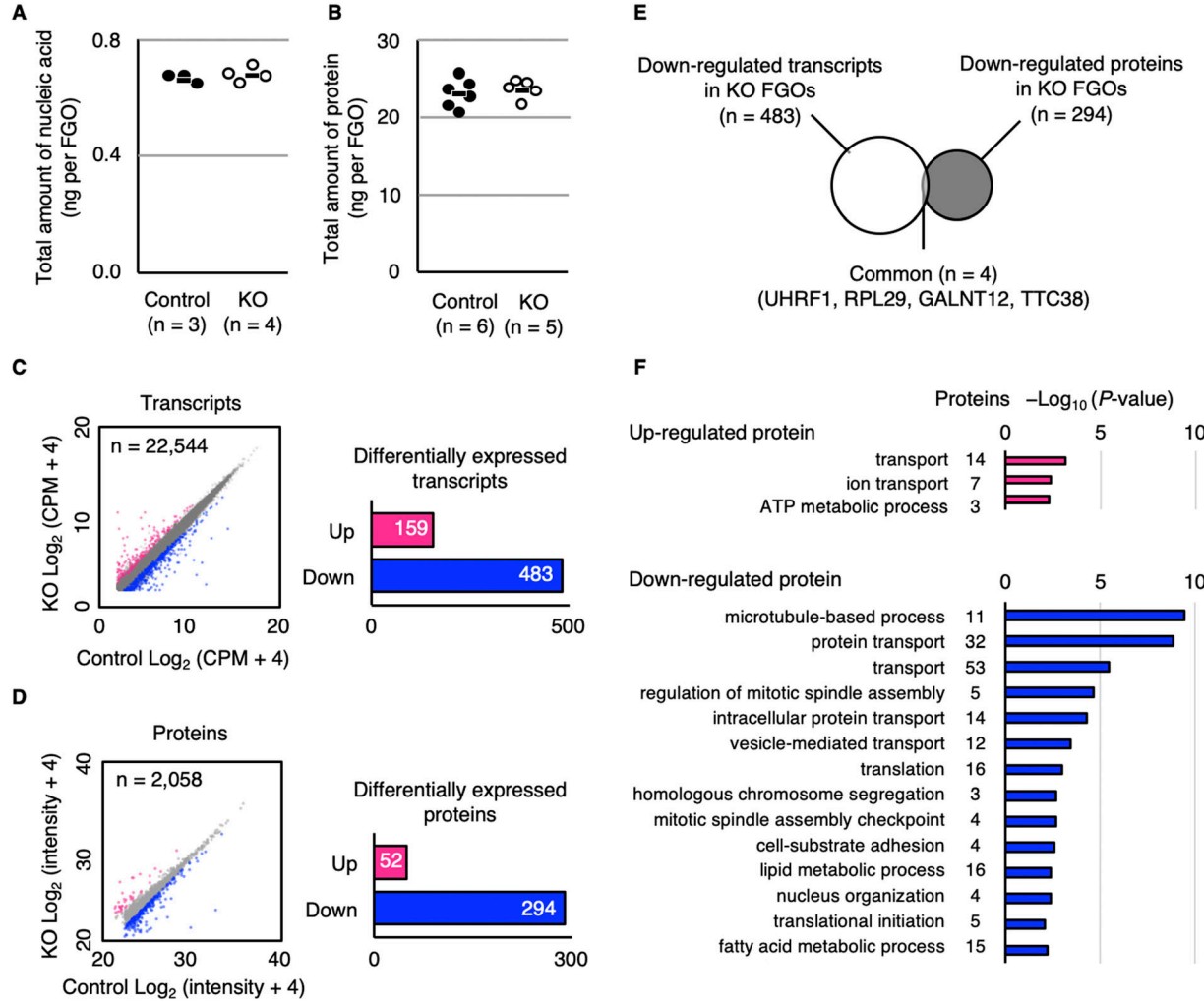

**Figure 3. UHRF1 is required for proper regulation of proteins associated with cytoskeletal organization.**
**(A)** Quantification of the total amount of nucleic acids in control [*Uhrf1*$^{2lox/2lox}$] and *Uhrf1* KO FGOs. Horizontal bars show the average values. Each dot represents one sample, which was a pool of 10 FGOs. **(B)** Quantification of the total amount of proteins in control and KO FGOs. Horizontal bars show the average values. Each dot represents one sample, which was a pool of 20–30 FGOs. **(C)** Scatter plot of transcripts expressed in control and KO FGOs (n = 22,544). Red and blue dots indicate up-regulated (≥2.0) and down-regulated (<0.5) transcripts, respectively. FDR < 0.01. CPM, count per million. The bar graph shows the number of differentially expressed transcripts in KO FGOs. **(D)** Scatter plot of proteins detected in control and KO FGOs (intensity ≥ 1 × 10$^7$, n = 2,058). Red and blue dots indicate up-regulated (≥2.0) and down-regulated (<0.5) proteins, respectively. The bar graph shows the number of up- and down-regulated proteins in KO FGOs. **(E)** Venn diagram illustrating the overlap of down-regulated transcripts (n = 483) and proteins (n = 294) in KO FGOs. Only four genes (proteins) overlapped. **(F)** Gene ontology analysis of the 52 up-regulated and 294 down-regulated proteins in KO FGOs.

examined the detailed transcriptomic and proteomic profiles of control [*Uhrf1*$^{2lox/2lox}$] and KO FGOs.

We previously reported that there was little change in the transcriptome of *Urhf1* KO FGOs (Maenohara et al, 2017), but others reported transcriptomic alterations in KO MII oocytes (Cao et al, 2019; Wu et al, 2020). We therefore revisited the transcriptome of KO FGOs using single-cell RNA sequencing (RNA-seq) (Ramskold et al, 2012). Hierarchical clustering of the transcriptomes of 10 control and 10 KO FGOs collected at 10–12 wk suggested genotype-dependent changes (Fig S1A): we identified 159 (0.73%) up-regulated (false discovery rate [FDR] < 0.01, fold change ≥ 2.0 [log$_2$ fold change ≥ 1.0]) and 483 (2.22%) down-regulated (FDR < 0.01, fold change < 0.5 [log$_2$ fold change < –1.0]) transcripts in KO FGOs (Fig 3C and Tables S1 and S2). A gene ontology (GO) analysis using the Database for Annotation, Visualization, and

Integrated Discovery (DAVID) (Sherman et al, 2022) revealed enrichment of these transcripts for some biological processes, including DNA methylation (Fig S1B).

For proteomic profiling, 50 FGOs were pooled (three batches for each genotype) and analyzed by liquid chromatography–tandem mass spectrometry (LC-MS/MS). Proteins that showed a maximum signal intensity (abundance) below 1 × 10$^7$ or those detected in only one replicate sample were not used for further analyses. A total of 2,058 proteins passed this filtration (Table S3), and we identified 52 (2.5%) up-regulated (log$_2$ fold change ≥ 1.0; Table S4) and 294 (14.3%) down-regulated (log$_2$ fold change < –1.0; Table S5) proteins in KO FGOs (Fig 3D). Changes were confirmed for selected proteins by Western blotting (Fig S2). Interestingly, we found a minimum overlap between the differentially expressed transcripts and proteins

(up-regulated, n = 0; down-regulated, n = 4 including UHRF1) (Fig 3E), suggesting that the changes in the proteome were independent of the transcriptome and likely occurred at the protein level.

A GO analysis revealed significantly enriched biological processes for the down-regulated proteins, including microtubule-based process ($P$ = 3.48 × $10^{-10}$), protein transport ($P$ = 1.35 × $10^{-9}$), and mitotic spindle assembly ($P$ = 2.19 × $10^{-5}$) (Fig 3F). For further characterization, we focused on proteins that are relatively abundant (top 20) in oocytes. Among such down-regulated proteins, six were tubulins (TUBA1C, TUBB2A, TUBB4A, TUBB4B, TUBB5, and TUBB6), which was consistent with the observed cytoplasmic defects and biological terms revealed by the GO analysis. Oocytes have specific microtubule meshwork called cytoplasmic lattices (CPLs) (Kan et al, 2011). Some members of the nucleotide-binding oligomerization domain, leucine-rich repeat, and pyrin domain–containing (NLRP) protein family are components of the subcortical maternal complex (SCMC), which is essential for CPL formation (Kim et al, 2010; Qin et al, 2019). We found that NLRP4A, NLRP9B, and NLRP14 were down-regulated (Table S5) although they are not known to be SCMC components. We therefore examined the levels of SCMC components and found that although not qualified as down-regulated ($\log_2$ fold change ≥ −1.0), NLRP5 and NLRP4F were less abundant in KO FGOs than control FGOs (fold change = 0.63 and 0.56 [$\log_2$ fold change = −0.66 and −0.83], respectively), along with other components of the SCMC (Tables S3 and S6). Ubiquitin-conjugating enzyme UBE2D3, an interacting partner of UHRF1 and reported to be necessary for spindle formation, chromosome segregation, and polar body extrusion (Ben-Eliezer et al, 2015), was also down-regulated (Table S5). Although the number of up-regulated proteins was small (Table S4) and their association with a specific biological process was less significant (Fig 3F), NUP85 and NUP37 are components of the Nup107-160 nuclear pore subcomplex presumed to be required for chromosome segregation (Orjalo et al, 2006). Proteins related to mitochondria (PMPCB, ETHE1, ATP5MG, and TMEM135) were markedly up-regulated, and those related to the actin cytoskeleton (CTTN and TLN1) were also up-regulated (Table S4). Taken together, many proteins related to microtubule-based processes and cytoplasmic functions were found to be misregulated in *Uhrf1* KO FGOs, which likely occurred independent of transcriptomic changes.

### UHRF1 is required for proper organization and localization of CPLs, organelles, and SCMC components

Based on the results of the proteomic analysis, we examined the localization of α-tubulin, a major component of CPLs, by immunostaining. Strikingly, CPLs were severely disorganized in *Uhrf1* KO FGOs, except in the subcortical region (Fig 4A). An electron microscopic analysis revealed a lack of CPLs in many regions of the cytoplasm (Fig 4B). We also observed mislocalization of organelles, such as mitochondria and ER, in KO FGOs (Fig 4C) and mat-KO two-cell embryos (Fig S3).

As the SCMC is essential for not only CPL formation but also organelle localization (Tong et al, 2000; Kim et al, 2010; Kan et al, 2011; Fernandes et al, 2012; Yu et al, 2014), and given that many SCMC components were down-regulated (Table S6), we performed an immunostaining analysis of the components. This analysis revealed

that two of components, NLRP5 and OOEP, changed their localization and showed a dispersed pattern (Fig 4D). It is interesting that *Nlrp5* and *Ooep* mat-KO embryos show cellular and developmental phenotypes very similar to those of *Uhrf1* mat-KO embryos (Tong et al, 2000; Esposito et al, 2007; Li et al, 2008; Kim et al, 2010; Tashiro et al, 2010). We suspect that these drastic changes in the architecture of oocyte cytoplasm are closely linked to the developmental phenotype.

## Discussion

It was previously reported that *Uhrf1* mat-KO embryos die before implantation (Maenohara et al, 2017; Cao et al, 2019), and a number of cytological and molecular defects seemingly associated with the lethal phenotype were described in KO oocytes and derived embryos. These defects include DNA hypomethylation, histone modification changes, transcriptomic changes, DNA damage, abnormal spindle formation, and aneuploidy (Maenohara et al, 2017; Cao et al, 2019). In the present study, we observed severely disorganized CPLs and mislocalized mitochondria and ER already at the FGO stage, in addition to the above-listed changes. The microtubule-associated changes are likely the primary cause of impaired chromosome segregation and abnormal cleavage division (this study), as well as the previously reported abnormal spindle shape and aneuploidy in MII oocytes (Cao et al, 2019). Consistent with this idea, our pronuclear transfer in zygotes showed that the major cause of the lethality resides in the cytoplasm, not the nucleus. This also suggests that the reported epigenetic defect carried by the maternal chromatin is not a serious problem for subsequent development.

The defects in the cytoplasmic architecture and microtubule-based processes can be explained by our proteomic profiling data, which showed the down-regulation of many proteins involved in CPL formation (e.g., tubulins and SCMC components) and microtubule-based processes (including transport and mitotic spindle assembly). Although the number of up-regulated proteins was smaller than the down-regulated proteins, some of them were also known to be associated with chromosome segregation, actin cytoskeleton, or mitochondria. The changes in these proteins together may have contributed to the lethality and phenotype of *Uhrf1* mat-KO embryos.

Regarding how the proteomic changes are brought about by the loss of UHRF1 in oocytes, although the pronuclear transfer findings suggest cytoplasmic defects to be the cause of the embryonic lethality, this does not necessarily exclude UHRF1's role in the nucleus. For example, epigenetic changes in oocytes may lead to altered gene expression, which can then affect the levels of mRNA and protein. However, although we detected transcriptomic changes specific to *Uhrf1* KO FGOs, there was little overlap between the transcriptomic and proteomic changes, suggesting that the proteomic changes linked to the phenotype occurred independently of the altered gene expression. Indeed, our GO analysis of the up- and down-regulated transcripts revealed the biological processes, which are basically irrelevant to the observed phenotype. It is therefore likely that the phenotype-linked proteomic changes occurred post-translationally in the cytoplasm of oocytes.

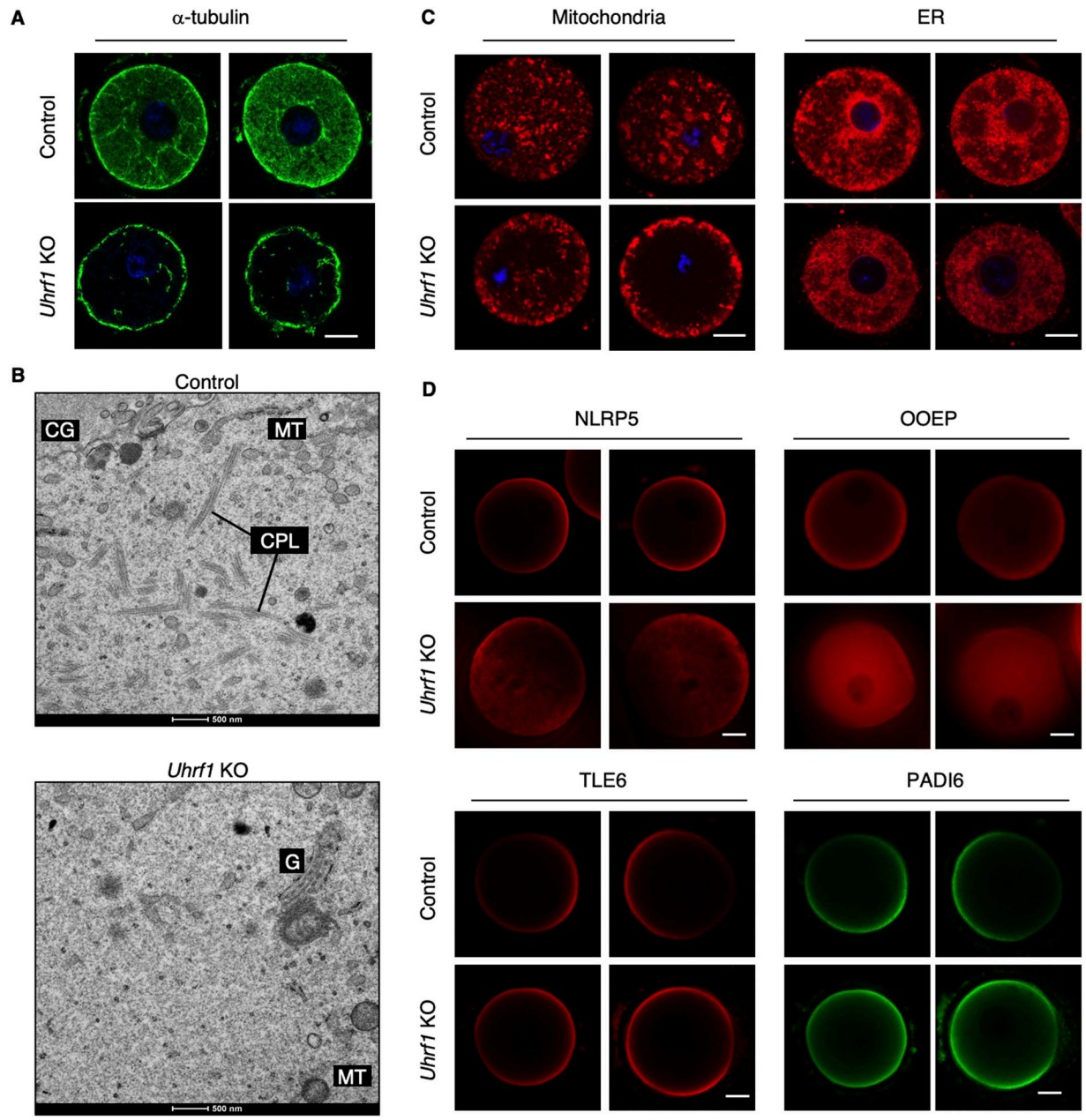

**Figure 4. UHRF1 is required for proper organization and localization of CPLs, mitochondria, and SCMC components.**
**(A)** Immunostaining of control [*Uhrf1*$^{2lox/2lox}$] and *Uhrf1* KO FGOs with anti-α-tubulin antibody (green; n = 12 and 25, respectively). DNA was stained with Hoechst 33342 (blue). Scale bar, 20 μm. **(B)** Representative transmission electron microscopic images of control and *Uhrf1* KO FGOs (n = 5 and 3, respectively). CPLs, cytoplasmic lattices; MT, mitochondria; CG, cortical granule; G, Golgi apparatus. Scale bar, 500 nm. **(C)** Staining of control and *Uhrf1* KO FGOs by MitoTracker (red; n = 10 and 9, respectively) and ER-Tracker (red; n = 31 and 31, respectively). DNA was stained with Hoechst 33342 (blue). Scale bar, 20 μm. **(D)** Immunostaining of control and *Uhrf1* KO FGOs with anti-NLRP5 antibody (red; n = 13 and 12, respectively), anti-OOEP antibody (red; n = 15 and 9, respectively), anti-TLE6 antibody (red; n = 13 and 16, respectively), or anti-PADI6 antibody (green; n = 13 and 16, respectively). Two representative images are shown for each condition. Scale bar, 20 μm.

At present, how cytoplasmic UHRF1 regulates proteins associated with CPL formation and microtubule-based processes is unknown. The protein has five distinct domains that can interact with other proteins (UBL domain, TTD, PHD, and SRA domain), recognize specific protein modifications (TTD), or catalyze protein ubiquitylation (UBL and RING domain). It is therefore tempting to speculate that UHRF1 serves as a hub for protein–protein interaction in the cytoplasm and, by doing so, regulates CPL assembly and microtubule-based processes. It has also become increasingly evident that UHRF1 has a variety of ubiquitylation targets, not only in the nucleus (histone H3 and PAF15) (Nishiyama et al, 2013, 2020), but also in the cytoplasm (STELLA, also known as DPPA3 or PGC7) (Funaki et al, 2014; Shin et al, 2017; Li et al, 2018b). Because protein ubiquitylation regulates many biological processes, including protein degradation (Rape, 2018), the RING domain of UHRF1 may play an important role in the cytoplasm of oocytes. To clarify which

function of UHRF1 is responsible for the phenotype, we are currently introducing mutations into each domain of UHRF1 and trying to observe the phenotypes.

In conclusion, we found that maternal UHRF1 regulates the proper cytoplasmic architecture and function of oocytes and preimplantation embryos, perhaps independently of its nuclear function, and ensures normal chromosome segregation and cell division in early development.

# Materials and Methods

## Ethics statement

Mouse husbandry and experiments were carried out in accordance with the ethical guidelines of Kyushu University, and the protocols were approved by the Animal Experiment Committee of Kyushu University.

## Animals

The generation of oocyte-specific *Uhrf1* KO mice [*Uhrf1*$^{2lox/2lox}$, *Zp3*-Cre] was described previously in Maenohara et al (2017). The mice were genotyped by polymerase chain reaction using the primers described previously in Maenohara et al (2017). All control and KO mice were of the C57BL/6J background (*Mus musculus domesticus*).

## Oocyte collection, in vitro fertilization (IVF), culture, and live-cell imaging

FGOs were collected from ovaries of female mice aged 8–12 wk. MII oocytes were obtained as cumulus–oocyte complexes from oviducts of ≥ 8-wk-old females injected with 7.5 U of pregnant mare serum gonadotropin and, 48 h later, with 7.5 U of human chorionic gonadotropin. For IVF, cumulus–oocyte complexes were incubated with C57BL/6J sperm. Cumulus cells were carefully removed by washing in PBS, and zygotes were cultured in potassium simplex optimized medium (KSOM) (EmbryoMax KSOM [1X] w/ 1/2 Amino Acids; Millipore Sigma) at 37°C with 5% $CO_2$. For live-cell imaging, zygotes were injected with 5 ng/$\mu$l mRNA encoding mCherry-tagged histone H2B and EGFP-tagged $\alpha$-tubulin at 5 h post-fertilization (Yamagata et al, 2009; Yamagata & Ueda, 2013). Imaging was performed for 94 h in a stage incubator on inverted microscopy (IX-71; Olympus Corp.) attached to a spinning-disk confocal unit (CSU-W1; Yokogawa Electric Corp). Raw images were stacked and projected by MetaMorph software (Molecular Devices) to generate reconstructed videos.

## Pronuclear transfer and embryo transfer

Donor and recipient zygotes generated by IVF were placed in a drop of Hepes-buffered KSOM containing 5 $\mu$g/ml cytochalasin B. The male and female pronuclei of a donor zygote were aspirated into a glass pipette (about 15 $\mu$m inner diameter) attached to a piezo-driven micromanipulator (Prime Tech). After exposed to the inactivated hemagglutinating virus of Japan (HVJ) (GenomONE-CF; Ishihara Sangyo), the pronuclei were inserted into the perivitelline space of a recipient zygote

that had been enucleated, as described above. Recipient zygotes were then cultured in fresh KSOM at 37.5°C under 5% $CO_2$ for about 30 min. The recipient zygotes fused with the pronuclei were washed with fresh KSOM and further cultured for 24 h. The next day, the reconstructed zygotes that had reached the two-cell stage were transferred into the oviducts of day-1 pseudopregnant ICR females.

## Total nucleic acid quantification

Ten FGOs were lysed by repeated freezing in liquid nitrogen and thawing in 2.5 $\mu$l of 1% SDS in PBS. Total nucleic acid quantification was performed using Quant-iT RiboGreen RNA Reagent (Thermo Fisher Scientific) according to the high-range assay approach. In brief, FGO lysates were diluted to 100 $\mu$l by Tris–EDTA (TE) buffer and 100 $\mu$l of 200-fold diluted RiboGreen reagent was added. Samples were incubated for several minutes at room temperature. The fluorescence of the sample was measured using EnSpire Multimode Plate Reader (PerkinElmer) and with standard fluorescein wavelengths (excitation 480 nm, emission 520 nm).

## Single-cell RNA-seq and data analyses

Total RNA was obtained from each FGO, and RNA-seq libraries were prepared using SMART-Seq Stranded Kit (Takara Bio) according to the standard protocol (Ramskold et al, 2012). In brief, total RNA was fragmented at 85°C for 6 min and then processed under the ultra-low-input workflow. PCR1 was performed for 10 cycles, and PCR2 was performed for 12 cycles. The final cleanup was performed twice. The libraries were sequenced on an Illumina NovaSeq 6000 using SP Reagent Kit (paired-end 151 nucleotides). Reads were trimmed and mapped to the reference mouse genome (mm10) by HISAT2 v2.1.0 (Kim et al, 2019). Transcripts were assembled by StringTie v2.1.3 (Kovaka et al, 2019). For hierarchical clustering and identification of the differentially expressed genes, iDEP online tools were used (Ge et al, 2018). Transcripts were filtered out by the criteria of at least 0.5 counts per million in all samples.

## Protein quantification

20–30 FGOs were lysed by repeated freezing in liquid nitrogen and thawing in 5 $\mu$l of 1% SDS in PBS. Protein quantification was performed using CBQCA Protein Quantification Kit (Thermo Fisher Scientific). In brief, FGO lysates were diluted to 120 $\mu$l with PBS, and then, 5 $\mu$l of 20 mM potassium cyanide (KCN) and 10 $\mu$l of 5 mM ATTO-TAG CBQCA reagent were added. Samples were incubated for 1 h at room temperature. The fluorescence of the sample was measured using EnSpire Multimode Plate Reader (PerkinElmer) with standard fluorescein wavelengths (excitation 465 nm, emission 550 nm).

## Proteomic profiling by LC–MS/MS

Fifty FGOs were lysed in 30 $\mu$l of 1 × Laemmli's sample buffer. After reduction with 10 mM TCEP at 100°C for 10 min and alkylation with 50 mM iodoacetamide at ambient temperature for 45 min, protein samples were subjected to SDS–PAGE. The electrophoresis was stopped at the migration distance of 2 mm from the top edge of the separation gel. After CBB staining, protein bands were excised,

destained, and cut finely before in-gel digestion with Trypsin/Lys-C Mix (Promega) at 37°C for 12 h. The resulting peptides were extracted from gel fragments and analyzed with an Orbitrap Fusion Lumos mass spectrometer (Thermo Fisher Scientific) combined with UltiMate 3000 RSLC nano-flow HPLC (Thermo Fisher Scientific). Peptides were enriched with $\mu$-Precolumn (0.3 mm i.d. x 5 mm, 5 $\mu$m; Thermo Fisher Scientific) and separated on an AURORA column (0.075 mm i.d. x 250 mm, 1.6 $\mu$m; Ion Opticks Pty Ltd) using the two-step gradient: 2–40% acetonitrile for 110 min, followed by 40–95% acetonitrile for 5 min in the presence of 0.1% formic acid. The analytical parameters of Orbitrap Fusion Lumos were set as follows: resolution of full scans = 50,000; scan range (m/z) = 350–1,500; maximum injection time of full scans = 50 msec; AGC target of full scans = $4 \times 10^5$; dynamic exclusion duration = 30 s; cycle time of data-dependent MS/MS acquisition = 2 s; activation type = HCD; detector of MS/MS = ion trap; maximum injection time of MS/MS = 35 msec; and AGC target of MS/MS = $1 \times 10^4$. The MS/MS spectra were searched against the *M. musculus* protein database in Swiss-Prot using Proteome Discoverer 2.4 software (Thermo Fisher Scientific), with an FDR < 1% set for peptide identification filters. Label-free relative quantification analysis for proteins was performed with the default parameters of the Minora Feature Detector node, Feature Mapper node, and Precursor Ions Quantifier node in Proteome Discoverer 2.4 software (described as relative protein abundance in Table S3). Finally, averaged label-free quantification intensities of biological triplicates were compared between control and *Uhrf1* KO FGOs. Replicates with zero intensity were excluded from the average calculation.

### GO analyses with DAVID

GO analyses were performed using DAVID (https://david.ncifcrf.gov/home.jsp), which provides a comprehensive set of functional annotation tools to understand the biological meaning behind a large list of genes or proteins (Sherman et al, 2022).

### Western blotting

Twenty FGOs were lysed in 10 $\mu$l of sample buffer (62.5 mM Tris–HCl [pH 6.8], 0.5 × PBS, 2% SDS, 10% glycerol, and 5% 2-mercaptoethanol). Proteins were denatured by heating at 95°C for 3 min, separated by electrophoresis on a 10% or 14% polyacrylamide gel, and transferred onto a PVDF membrane (Bio-Rad). The blots were blocked with 5% skimmed milk and incubated with primary antibodies against FBXO38 (1:1,000, ab87729; Abcam), UBE2D3 (1:1,000, 4330; Cell Signaling Technology), or SPIN1 (1:1,000, 12105-1-AP; Proteintech). After washing several times, the blots were incubated with HRP-conjugated anti-rabbit IgG (1:30,000, ab6721; Abcam) as the secondary antibody, and proteins were detected using the Chemi-Lumi One Ultra reagent (11644-40; Nacalai Tesque) and an LAS-3000 Lumino-image analyzer (Fujifilm).

### Immunostaining

Immunostaining was performed as described previously with minor modifications (Yurttas et al, 2008; Yu et al, 2014; Maenohara et al, 2017). For $\alpha$-tubulin staining, FGOs were firstly permeabilized with 0.2% Triton X-100 in PBS and then fixed with 4% PFA. The cells were incubated with the antibody against $\alpha$-tubulin (1:200, 2125; Cell Signaling Technology) at 4°C overnight. After washing several times, the cells were incubated with CF488A-conjugated anti-rabbit IgG (1:1,000, 20015; Biotium) and Hoechst 33342 (1:1,000) for 30 min at room temperature. The cells were washed with PBS and observed using LSM700 confocal laser scanning microscope (Carl Zeiss). For the detection of other proteins, FGOs were fixed with 4% PFA and permeabilized with 0.5% Triton X-100 in PBS for 30 min at room temperature. The cells were then incubated at 4°C overnight with the antibodies against NLRP5 (1:200, kind gift from Scott A. Coonrod), OOEP (1:200, PA5-85954; Thermo Fisher Scientific), PADI6 (1:200, kind gift from Scott A. Coonrod), or TLE6 (1:200, kind gift from Jurrien Dean) at 4°C overnight. After washing several times, the cells were incubated with CF594-conjugated anti-rabbit IgG (1:1,000, 20152; Biotium) or CF488A-conjugated anti-guinea pig IgG (1:1,000, 20169; Biotium) for 30 min at room temperature. After mounting in VEC-TASHIELD medium with DAPI (Vector Laboratory), the cells were observed using LSM700. For visualization of organelles, FGOs were incubated with 1 $\mu$M MitoTracker Red (M7512; Thermo Fisher Scientific) in M2 medium (Sigma-Aldrich), 1 $\mu$M MitoTracker Green FM (M7514; Thermo Fisher Scientific) in KSOM medium (Millipore), or 1 $\mu$M ER-Tracker Red (E34250; Thermo Fisher Scientific) in KSOM medium, each containing Hoechst 33342 (1:1,000), for 1 h at 37°C with 5% $CO_2$. After transferring to M2 or KSOM medium without dyes, the cells were observed using LSM700.

### Transmission electron microscopy

Imaging using a transmission electron microscope was performed as described previously with minor modifications (Gotoh et al, 2018). Ovaries were immersed in 2.5% glutaraldehyde in 0.1 M cacodylate buffer at room temperature for 2 h. Samples were post-fixed with 1% $OsO_4$ in 0.1 M sucrose buffer at 4°C for 2 h. Tissue samples were dehydrated in a graded series of ethanol washes. Ultrathin sections were prepared with an ultramicrotome (EM UC7; Leica) and stained with 2% uranyl acetate and lead citrate. Sections were visualized using a transmission electron microscope (Tecnai 20; FEI Co).

## Data Availability

The sequence datasets supporting the results of this article are available in the DDBJ Sequence Read Archive under the accession number DRA013956.

## Supplementary Information

## Acknowledgements

We thank Rao Huo (Nanjing Medical University) for raw mRNA-seq data of *Uhrf1* KO MII oocytes, Scott A. Coonrod (Baker Institute for Animal Health) for anti-NLRP5 and anti-PADI6 antibodies, and Jurrien Dean (National Institutes

of Health) for anti-NLRP5, anti-TLE6, and anti-KHDC3 antibodies. We also thank Kyohei Arita (Yokohama City University) for his valuable advice, Ryo Ugawa (Laboratory for Research Support, Medical Institute of Bioregulation, Kyushu University) for performing transmission electron microscopy, and Tomoko Hanagiri, Miho Miyake, Tomomi Akinaga, and Junko Oishi (Kyushu University) for their technical assistance. This work was supported by JSPS KAKENHI grant numbers JP18H05214 (to H Sasaki), JP15K06803, JP24613005, and JP19H05740 (to M Unoki), JP19H05758 (to A Ogura), JP25712035 and JP25116005 (to K Yamagata), and a grant from Naito Foundation (to M Unoki).

## Author Contributions

S Uemura: formal analysis, investigation, and writing—original draft.
S Maenohara: formal analysis and investigation.
K Inoue: formal analysis, investigation, and methodology.
N Ogonuki: formal analysis, investigation, and methodology.
S Matoba: formal analysis, investigation, and methodology.
A Ogura: formal analysis, supervision, investigation, and methodology.
M Kurumizaka: formal analysis, investigation, and methodology.
K Yamagata: formal analysis, investigation, and methodology.
J Sharif: resources, formal analysis, and investigation.
H Koseki: resources and supervision.
K Ueda: formal analysis, investigation, and methodology.
M Unoki: conceptualization, formal analysis, supervision, funding acquisition, investigation, and writing—original draft.
H Sasaki: conceptualization, formal analysis, supervision, funding acquisition, investigation, and writing—original draft, review, and editing.

## Conflict of Interest Statement

The authors declare that they have no conflict of interest.

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
