## [Reviewer comments · Life Science Alliance]

Life Science Alliance

UHRF1 is essential for proper cytoplasm architecture and function of mouse oocyte and derived embryo

Shuhei Uemura, Shoji Maenohara, Kimiko Inoue, Narumi Ogonuki, Shogo Matoba, Atsuo Ogura, Mayuko Kurumizaka, Kazuo Yamagata, Jafar Sharif, Haruhiko Koseki, Koji Ueda, Motoko Unoki, and Hiroyuki Sasaki

DOI: <https://doi.org/10.26508/lsa.202301904>

Corresponding author(s): Hiroyuki Sasaki, Kyushu University

Review Timeline:

Submission Date:	2023-01-06
Editorial Decision:	2023-02-02
Revision Received:	2023-04-27
Editorial Decision:	2023-05-12
Revision Received:	2023-05-14
Accepted:	2023-05-15

Transaction Report:

February 2, 2023

Re: Life Science Alliance manuscript #LSA-2023-01904-T

Prof. Hiroyuki Sasaki
Kyushu University
3-1-1 Maidashi
Higashi-ku
Fukuoka 812-8582

Dear Dr. Sasaki,

Thank you for submitting your manuscript entitled "UHRF1 is essential for proper cytoplasmic architecture and function of mouse oocytes and derived embryos" to Life Science Alliance. The manuscript was assessed by expert reviewers, whose comments are appended to this letter. We invite you to submit a revised manuscript addressing the Reviewer comments.

Thank you for this interesting contribution to Life Science Alliance. We are looking forward to receiving your revised manuscript.

Sincerely,

B. MANUSCRIPT ORGANIZATION AND FORMATTING:

Reviewer #1 (Comments to the Authors (Required)):

In the manuscript "UHRF1 is essential for proper cytoplasmic architecture and function of mouse oocytes and derived embryos" the authors revisit the previously described UHRF1 oocyte-specific KO model and claim that the phenotypes of poor oocyte and embryos results solely due to cytoplasmic rather than nuclear defects.

Fig.1 compares the developmental defects of mutant and control zygotes and describes various defects in KO oocyte-derived zygotes: Please provide quantification for uneven cell division, cytoplasmic fragmentation, misaligned chromosomes, lagging chromosomes, and micronuclei formation.

Fig. 2: The embryo survival data of transferred nuclear KO vs cytoplasmic KO is interesting, however, the number of embryos examined is very low. How many transfers were done to achieve this embryo number? 15/15 cytoplasmic KO embryos died. Were there any implants (dead or resorption sites)? Frankly, I am very uncomfortable with the number of embryos analyzed to ascertain the cytoplasmic abnormality as the sole cause of problem in the mutant embryos. It appears that the analysis of E19.5 embryos included only one embryo transfer experiment. I am not doubting the researchers but it is very difficult to obtain any conclusion by applying statistical methods on such a small data set. Since the conclusion of the paper is based on these embryo transfer data, I suggest either just reporting the transcriptomic and proteomic data in WT and KO FGOs in the paper or performing additional embryo transfers to get a conclusion on relative contribution of cytoplasm or nucleus. I believe that the authors claims are not very well supported by the current data.

Fig. 3: The analysis of gene expression is interesting and is performed well.

Fig. 4 results show cytoplasmic architecture defects in FGOs, however, there is no such data provided for the derived embryos as mentioned in the title of the manuscript. The embryos are expected to have cytoplasmic defects but the current results do not show this.

Reviewer #2 (Comments to the Authors (Required)):

While the nuclear functions of the UHRF1 protein (DNA methylation) are well appreciated (it appears), its cytoplasm roles in the early mouse (mammalian) oocyte, egg, and pre-implantation embryo have been less well studied. Here the authors use oocytes, eggs, embryos, and embryonic nuclei and enucleated cytoplasm derived from maternal UHRF1 knockout and wild type embryos in a range of studies that confirm and extend the observation that cytoplasmic UHRF1 plays an essential role in the early mouse embryo, and is involved in normal cytoplasmic organization, with significant functional impacts on membrane rigidity, mitotic and cytokinetic processes, as well as general aspects of cytoplasmic organization, including protein stability (from comparative RNA SEQ and proteomic studies), and the organization and integrity of subcortical maternal complexes and cytoplasmic lattices. An interesting observation is that proteomic and RNA SEQ studies revealed "a minimum overlap between the differentially expressed transcripts and proteins" suggesting cytoplasmic effects on protein stability or RNA translation. This confirms (a good thing) that UHRF1 acts cytoplasmically and seems well done and convincing.

It is a more general editorial decision whether it meets the impact standards of LSA. The obvious follow up study would be to exploit the resources and methods used here, together with site-directed mutagenesis to determine which of the domains / activities of the UHRF1 protein are involved in its cytoplasmic effects / roles, as the protein contains ubiquitin-like (UBL), a tandem Tudor, PHD, SET and RING associated, and a RING finger domains, but again, this is an editorial decision.

Point-by-point Response (LSA-2023-01904-T)

We appreciate the time and effort that the reviewers put into reviewing our manuscript and find their comments helpful to improve our manuscript. According to their suggestions, we have performed additional experiments and analyses, the outcomes of which further support our conclusions.

To accommodate the new data, we have revised **Figure 1B,C** and **Figure 2B**, their legends, and relevant portions of the main text. Below you will find our point-by-point response to the reviewers. All changes are **highlighted in red** in the revised manuscript.

Reviewer #1 (Comments to the Authors):

In the manuscript "UHRF1 is essential for proper cytoplasmic architecture and function of mouse oocytes and derived embryos" the authors revisit the previously described UHRF1 oocyte-specific KO model and claim that the phenotypes of poor oocyte and embryos results solely due to cytoplasmic rather than nuclear defects.

Our response: While we believe that nuclear defects contribute little to the developmental phenotype, we did not say that it is "solely" due to cytoplasmic. It is difficult to exclude contribution by the nuclear defects completely, and we therefore stated in our original manuscript that "the major cause of the preimplantation lethality resides in the cytoplasm" (**page 7, paragraph 1**). We have also added "major" to a similar expression in **Discussion**.

Fig.1 compares the developmental defects of mutant and control zygotes and describes various defects in KO oocyte-derived zygotes: Please provide quantification for uneven cell division, cytoplasmic fragmentation, misaligned chromosomes, lagging chromosomes, and micronuclei formation.

Our response: We thank the reviewer for this comment. To quantify each defect, we have carefully observed the movies (a total of five; an example is posted in **Supplemental Movie S1**), traced the developmental phenotype of *Uhrf1* mat-KO (n = 51) and control embryos (n = 57), and examined what fraction of the observed embryos show that defect. The results are indicated below the pictures of **Figure 1B,C**, and a clear difference is evident between the genotypes.

Fig. 2: The embryo survival data of transferred nuclear KO vs cytoplasmic KO is interesting, however, the number of embryos examined is very low. How many transfers were done to achieve this embryo number? 15/15 cytoplasmic KO embryos died. Were there any implants

(dead or resorption sites)? Frankly, I am very uncomfortable with the number of embryos analyzed to ascertain the cytoplasmic abnormality as the sole cause of problem in the mutant embryos. It appears that the analysis of E19.5 embryos included only one embryo transfer experiment. I am not doubting the researchers but it is very difficult to obtain any conclusion by applying statistical methods on such a small data set. Since the conclusion of the paper is based on these embryo transfer data, I suggest either just reporting the transcriptomic and proteomic data in WT and KO FGOs in the paper or performing additional embryo transfers to get a conclusion on relative contribution of cytoplasm or nucleus. I believe that the authors claims are not very well supported by the current data.

Our response: This reviewer concerns the reproducibility of the survival data of transferred nuclear KO and cytoplasmic KO embryos because of the small number (two) of embryo transfer experiment done for each pronuclei/cytoplasm constitution. He/she also wonders whether there were any implants (dead or resorption sites), which we clearly provided in parenthesis in our **original Figure 2B**. According to this reviewer's suggestion, we have performed four additional embryo transfer experiments respectively for nuclear KO and cytoplasmic KO embryos. Together with the data from the original manuscript, we now present data from a total of six transfer experiments for each pronuclei/cytoplasm constitution, of which four were examined at E12.5 and two at E19.5 (**new Figure 2B**). These data strongly support our original conclusion. We describe these results in **paragraph 1, page 7**.

Fig. 3: The analysis of gene expression is interesting and is performed well.

Our response: We thank the reviewer for this comment.

Fig. 4 results show cytoplasmic architecture defects in FGOs, however, there is no such data provided for the derived embryos as mentioned in the title of the manuscript. The embryos are expected to have cytoplasmic defects but the current results do not show this.

Our response: In addition to the uneven cleavage divisions and cytoplasmic fragmentation in mat-KO embryos (Figure 1), we showed the extraordinarily high mobility of cytoplasmic granules in mat-KO zygotes (Supplemental Movie S2), which we believe is a good indication of impaired cytoplasmic architecture and function. To further support the cytoplasmic defects, we have now stained mitochondria and ER in mat-KO 2-cell embryos and observed their clear mis-localization (new **Supplemental Figure S3**). We have also tried immunostaining for tubulins and SCMC components, but mat-KO embryos were so fragile that our repeated attempt was not successful. We are therefore continuing our effort to establish the condition. Anyway, we believe that our data well support the cytoplasmic

defects in embryos and hope that this reviewer will understand.

Reviewer #2 (Comments to the Authors):

While the nuclear functions of the UHRF1 protein (DNA methylation) are well appreciated (it appears), its cytoplasm roles in the early mouse (mammalian) oocyte, egg, and pre-implantation embryo have been less well studied. Here the authors use oocytes, eggs, embryos, and embryonic nuclei and enucleated cytoplasm derived from maternal UHRF1 knockout and wild type embryos in a range of studies that confirm and extend the observation that cytoplasmic UHRF1 plays an essential role in the early mouse embryo, and is involved in normal cytoplasmic organization, with significant functional impacts on membrane rigidity, mitotic and cytokinetic processes, as well as general aspects of cytoplasmic organization, including protein stability (from comparative RNA SEQ and proteomic studies), and the organization and integrity of subcortical maternal complexes and cytoplasmic lattices. An interesting observation is that proteomic and RNA SEQ studies revealed "a minimum overlap between the differentially expressed transcripts and proteins" suggesting cytoplasmic effects on protein stability or RNA translation. This confirms (a good thing) that UHRF1 acts cytoplasmically and seems well done and convincing.

Our response: We are happy to hear the positive comments from this reviewer.

It is a more general editorial decision whether it meets the impact standards of LSA. The obvious follow up study would be to exploit the resources and methods used here, together with site-directed mutagenesis to determine which of the domains / activities of the UHRF1 protein are involved in its cytoplasmic effects / roles, as the protein contains ubiquitin-like (UBL), a tandem Tudor, PHD, SET and RING associated, and a RING finger domains, but again, this is an editorial decision.

Our response: Thank you very much for this comment. We are indeed doing such experiments to determine which domain/function of the UHRF1 protein is responsible for the cytoplasmic phenotype. As one can easily imagine, however, production and analysis of such a series of mutant mice take time, and we hope that we can report the results of the mutagenesis study in a not-so-distant future.

May 12, 2023

RE: Life Science Alliance Manuscript #LSA-2023-01904-TR

Prof. Hiroyuki Sasaki
Kyushu University
3-1-1 Maidashi
Higashi-ku
Fukuoka 812-8582
Japan

Dear Dr. Sasaki,

Thank you for submitting your revised manuscript entitled "UHRF1 is essential for proper cytoplasm architecture and function of mouse oocyte and derived embryo". We would be happy to publish your paper in Life Science Alliance pending final revisions necessary to meet our formatting guidelines.

- please add a callout for Figure 2C in the text
- please add a scale bar to Figure 1A

A. FINAL FILES:

B. MANUSCRIPT ORGANIZATION AND FORMATTING:

****It is Life Science Alliance policy that if requested, original data images must be made available to the editors. Failure to provide**

original images upon request will result in unavoidable delays in publication. Please ensure that you have access to all original data images prior to final submission.**

The license to publish form must be signed before your manuscript can be sent to production. A link to the electronic license to publish form will be sent to the corresponding author only. Please take a moment to check your funder requirements.

Sincerely,

Reviewer #1 (Comments to the Authors (Required)):

I thank the authors for addressing my queries. I believe that the manuscript is much improved and I recommend for the publication of this manuscript.

May 15, 2023

RE: Life Science Alliance Manuscript #LSA-2023-01904-TRR

Prof. Hiroyuki Sasaki
Kyushu University
3-1-1 Maidashi
Higashi-ku
Fukuoka 812-8582
Japan

Dear Dr. Sasaki,

Thank you for submitting your Research Article entitled "UHRF1 is essential for proper cytoplasm architecture and function of mouse oocyte and derived embryo". It is a pleasure to let you know that your manuscript is now accepted for publication in Life Science Alliance. Congratulations on this interesting work.

DISTRIBUTION OF MATERIALS:

Again, congratulations on a very nice paper. I hope you found the review process to be constructive and are pleased with how the manuscript was handled editorially. We look forward to future exciting submissions from your lab.

Sincerely,
